# Evaluation of the National cervical cancer surveillance program in Bangladesh: Performance, strengths, and opportunities for improvement

Md Foyjul Islam[1]*, Ashrafun Nessa[2], Quazi Ahmed Zaki[1], Shah Ali Akbar Ashrafi[3], Md. Omar Qayum[1], Mohammad Rashedul Hassan[1], Tahmina Shirin[1]

1 Institute of Epidemiology, Disease Control and Research (IEDCR), Dhaka, Bangladesh, 2 Bangladesh Medical University (BMU), Dhaka, Bangladesh, 3 Mangaement Information System, Director General of Health Services, Bangladesh

* drislam0666@gmail.com

## Abstract

Cervical cancer (CC) is the second most prevalent cancer among women in Bangladesh, with 9,640 new cases and 5,826 deaths annually. Since 2018, the Electronic Data Tracking with Population-Based Cervical and Breast Cancer Screening Program (EPCBCSP) has been implemented nationwide across 601 health centers, utilizing Visual Inspection with Acetic Acid (VIA) for CC screening. However, the system had not been evaluated until now. This study aims to assess the performance of EPCBCSP and identify its strengths and challenges. We evaluated EPCBCSP using the updated guidelines from the Centers for Disease Control and Prevention(CDC) in 2023. Data were collected from central level stakeholders and from stakeholders at one colposcopy and one VIA center in each of four selected divisions of Bangladesh. Interviews with 42 stakeholders assessed simplicity, flexibility, acceptability, stability and usefulness attributes. Data quality was assessed by surveillance data review. Percentage scores were calculated and ranked as follows: excellent (≥ 90%), Very good (≥ 80%-<90%.), good (≥60%-<80%), average (≥50%-<60%) and poor (≤50%).The overall system performance was rated as good (score:77.20%). The system's simplicity (score: 91.61%) and usefulness (score:91.61%) were ranked as "excellent". The acceptability (score: 85.29%) was ranked as "very good and flexibility (score:71.90%) was rated as "good". Stability (percentage score: 53.25%) was considered average. Furthermore, data quality was assessed as average based on missing values (13.11%.) Electronic data tracking system, instant test report availability and "see and treat" policies were identified as strengths by stakeholders, while dedicated staff shortages and motivating participants were main challenges. The evaluation EPCBCSP in Bangladesh indicates overall good performance, excelling in simplicity and usefulness. However, improvements are needed in stability and

**Data availability statement:** The data collected for this study has been analyzed and is included within this published article. The dataset is attached as supplementary information.

**Funding:** The authors received no specific funding for this work. However, the study was partially supported by the Institute of Epidemiology, Disease Control and Research (IEDCR) as part of the routine academic activities of the Field Epidemiology Training Program (FETP), Bangladesh. The funders had no role in study design, data collection and analysis, decision to publish, or preparation of the manuscript.

**Competing interests:** The authors have declared that no competing interests exist.

data quality. Strengthening surveillance efforts addressing shortages of dedicated staff and participant motivation will further enhance the effectiveness of the program, crucial for reducing CC burden.

## Introduction

Cervical cancer (CC) is primarily caused by persistent infection with high-risk human papillomavirus (HPV), a common sexually transmitted infection responsible for 99.7% of cases [1]. Cervical cancer was the fourth most common cancer in women globally, with approximately 660,000 new cases and 350,000 deaths in 2022, 94% of which occurred in low- and middle-income countries [2–4].In Bangladesh, cervical cancer is the second most common cancer among women, accounting for 13.3% of all female cancers, with a five-year case count of 26,698 and a prevalence rate of 32.1 per 100,000 population [5]. In 2022, there were 9,640 new cases and 5,826 deaths reported, with incidence and mortality rates significantly higher than the global averages (incidence: 5.8 vs. 3.3 per 100,000 women; mortality: 5.0 vs. 3.6 per 100,000 women) [5,6].The overall 5-year relative survival rate for cervical cancer is 73.2% for early and localized stages but falls to 7.4% for advanced stages; however, in Bangladesh, the high mortality rates are largely due to late-stage diagnoses and inadequate management facilities [7,8]. The global strategy to eliminate cervical cancer by 2030 includes achieving an incidence rate of 4 per 100,000 women-years, vaccinating 90% of girls by age 15, screening 70% of women by ages 35 and 45, and treating 90% of women with cervical disease.

To eliminate cervical cancer, Bangladesh initiated a Visual Inspection with Acetic Acid (VIA)-based national cervical cancer screening program in 2004, aiming to detect precancerous lesions early and reduce the disease burden. The VIA method was chosen for its simplicity, cost-effectiveness, and applicability in low-resource settings. Despite these efforts, challenges remain in achieving the desired coverage and quality of care [9,10].

The National Cervical Cancer Surveillance System (NCCSS) was initiated in Bangladesh by a pilot program in 2005 at 16 District Hospitals (DHs), 16 Maternal and Child Welfare Centers (MCWCs), and 12 Union Health and Family Welfare Centers (UH & FWCs) with technical assistance from UNFPA [9,11]. After the Pilot Program was successfully completed, the Ministry of Health and Family Welfare (MOHFW) decided to expand the program to include the remaining district level health infrastructure (MCHs, DHs, and MCWCs) and incorporate clinical breast examination (CBE) for breast cancer screening. All ever-married women over the age of 30 were made up the target group for CC screening, and these women were motivated to use VIA and CBE whenever they visited the facilities for various reasons [12]. The National Strategy for Cervical Cancer Prevention & Control (2017–2022) was developed by specialist group represented by stakeholders from government and NGO's and development partners and approved by Ministry of Health and Family Welfare(MOHFW), Bangladesh [13]. The overall objective of this National Strategy was

to guide, develop, strengthen strategies to improve cervical cancer control activities; to reduce the burden of morbidity, disability and death from cervical cancer and to promote women's good health. The GOB planned to implement organized population-based cancer screening program through the public health delivery system to achieve a reasonable coverage of the target population [7,14]. Since 2018, surveillance has been conducted under the "Electronic Data Tracking with Population-Based Cervical and Breast Cancer Screening Program" across 601 health facilities encompassing both urban, semi-urban, and rural populations covering all sub districts in Bangladesh. All facilities offer VIA screening test, with 43 centers also equipped with colposcopy facilities. Data were collected by online platform of District Health Information System 2(DHIS2) through Management information system (MIS). The assigned person can input data to the system by login through user Id and password. When a respondent register for screening (in Community Clinic by CHCP, in screening facilities by assigned SSN) a unique identification number was generated. Respondents information was inputted to the system during registration. Screening result is updated by facility assigned staff [10,12,15] **[Fig 1]**.

The NCCSS is vital for monitoring the screening program's effectiveness, identifying service gaps, and informing policy decisions. However, challenges like limited public awareness, inadequate infrastructure, and unequal access to care persist, particularly in rural areas. Despite its importance, the system has not been evaluated since its launch. This study aims to assess the system's performance using CDC indicators—simplicity, flexibility, acceptability, stability, data quality, and usefulness—while identifying strengths, weaknesses, and opportunities, and providing recommendations for improvement.

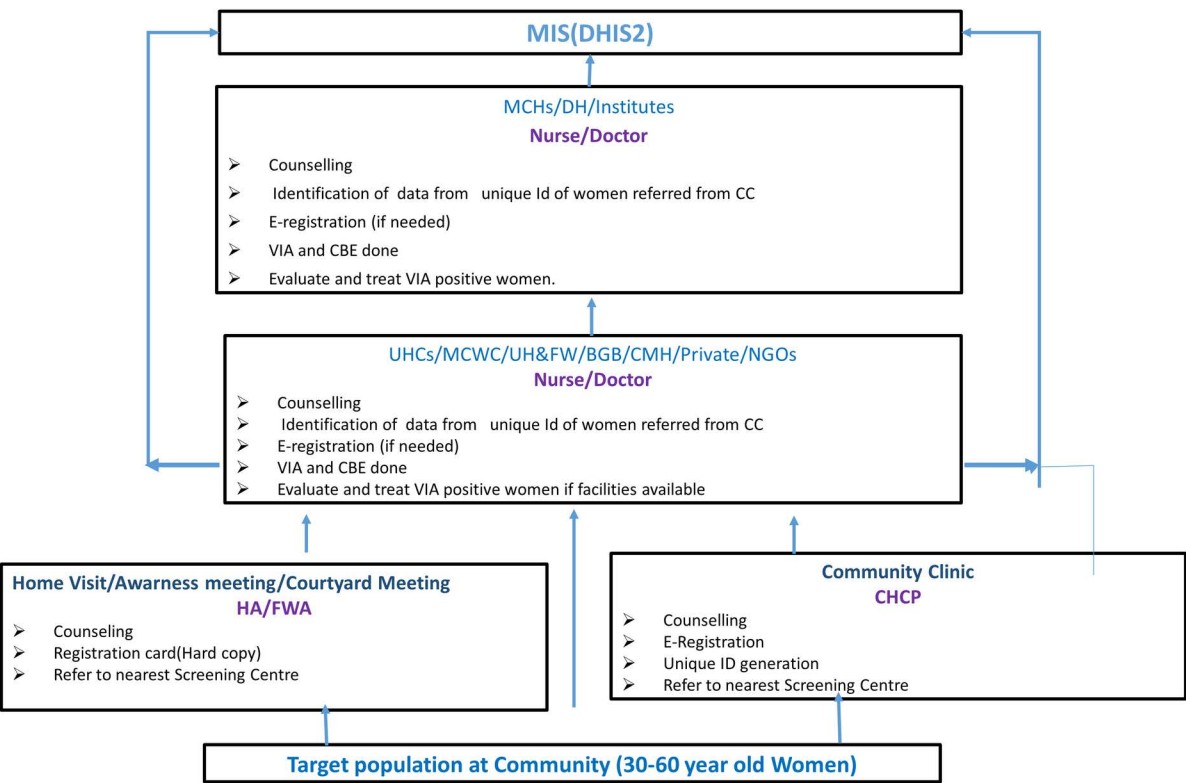

**Fig 1. Flow of data in the reporting system of cervical cancer surveillance system.**

## 2.0 Methodlogy

### Study design

A cross-sectional descriptive evaluation study using mixed quantitative and qualitative methods was conducted based on the updated Centers for Disease Control and Prevention guidelines for the evaluation of a public health surveillance system to assess the NCCSS from February 2022 to July 2023.The number of stakeholders was selected based on the purpose, timeline and resources [16].

### Study population and sampling

The study involved 42 stakeholders, including two central-level participants: Project Director of the EPCBCSP and the Chief of the Health Information Unit (HDU), Management Information System (MIS), DGHS. Eight study units/sites were selected from four randomly chosen divisions (Dhaka, Khulna, Sylhet, Rajshahi), including four VIA centers and four colposcopy centers, one from each division. [Fig 2]

The participants from these centers included three hospital directors, four UHFPOs, and one superintendent from a district hospital. Additionally, interviews were conducted with 13 nurses, six colposcopists, seven Community Healthcare Providers (CHCPs), and six Health Assistants.

### Data collection

Data were collected via in-depth interview using a semi-structured questionnaire [S1 Text] adapted from updated CDC guideline for evaluating public health surveillance system through key stakeholder interviews and review of all documents, guidelines, strategies, and pertinent scientific literature on NCCSS. All the stakeholder were interviewed at their workplace. The investigator himself was the data collector to minimize the subjectivity of responses from stakeholders. As we analyzed anonymous data, no identification of the patients was present, so there was no ethical issue. Data from 1st January 2018–31st December 2022 was obtained from DHIS2 on 23rd January 2023, with permission from the EPCBCSP authority, to assess data quality. Furthermore, all the confidential data were kept securely and handled appropriately.

### System attributes evaluation

A total of six surveillance system attributes that can affect surveillance were assessed. Quantitative analysis was used to assess data quality, while qualitative analysis was used to assess simplicity, flexibility, acceptability, and stability and Usefulness.

### Scoring system and interpretation

The interview responses were analyzed by calculating the number and percentage of respondents for fixed-option questions and summarizing free-text responses narratively. Data analysis was conducted using Microsoft Excel and a scientific calculator, with descriptive statistics applied to quantitative variables and summarized narratives for qualitative data.

With regard to qualitative attributes (simplicity, flexibility, acceptability, and stability, and usefulness) assessed using questions with yes (1) or no (0) answers and stakeholders were asked to rate the degree to which they agreed with attributes' specific indicators by using a 5-point Likert scale (**1=strongly disagree; 2=disagree; 3=neutral; 4=agree; 5=strongly agree)** and useful assessed by **(1=Not at all Useful at all; 2=not very Useful; 3=neutral; 4=Somewhat Usefulness; 5=Very Useful)**. Higher scores indicated better performance in terms of the studied attribute. Data quality was assessed by percentage of missing data using below formula.

$$Percedntage\ of\ missing\ data = \frac{Number\ of\ missing\ data\ fields}{Total\ expected\ data\ fields} x\ 100$$

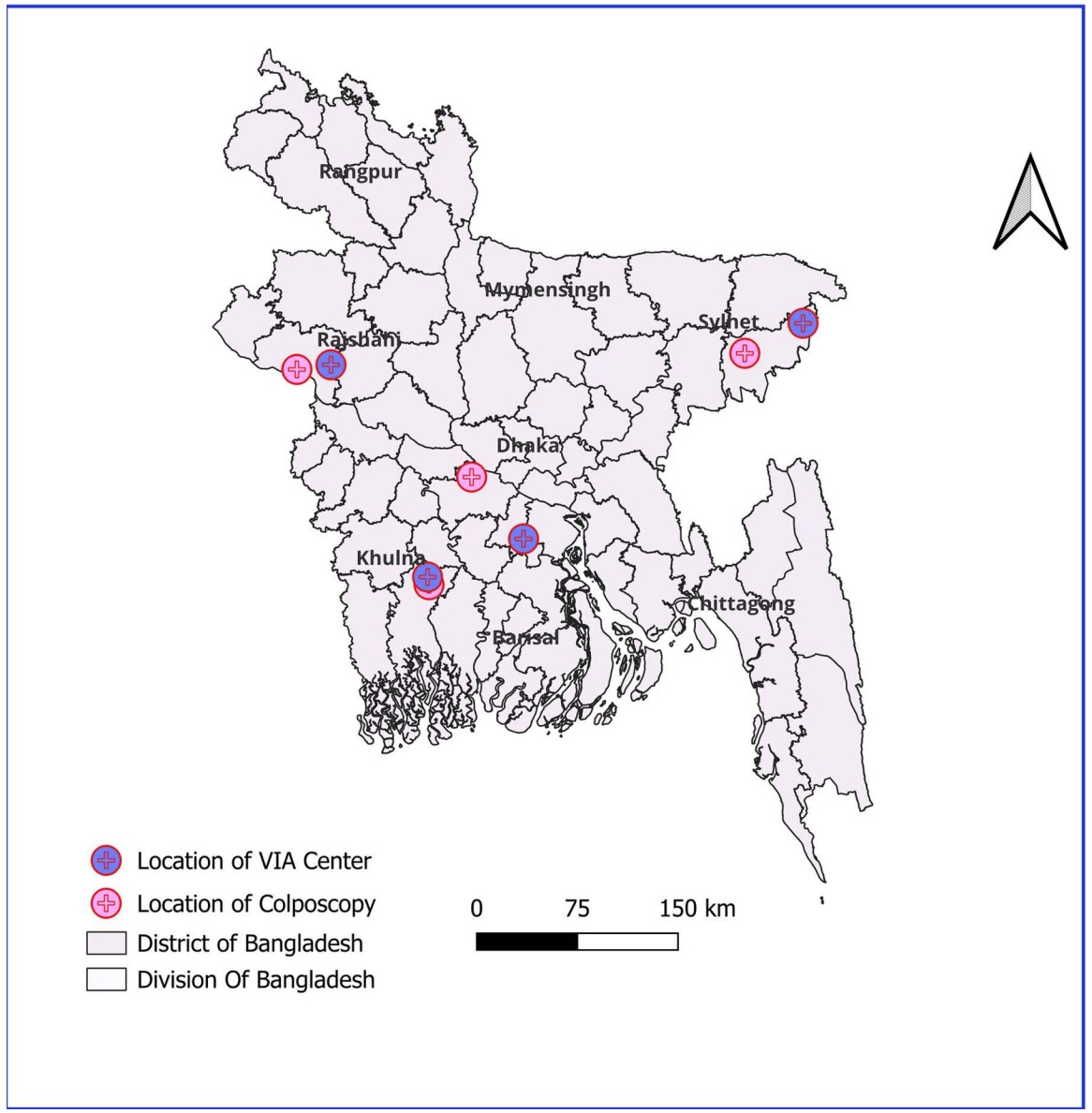

**Fig 2. Map showing selected VIA and Colposcopy center for study. Map created using QGIS 3.34. Administrative boundaries for districts of Bangladesh provided by GADM, version 4.1, under a Creative Commons Attribution license (https://gadm.org/download_country. html#google_vignette).**

This percentage was then categorized as Excelent=<2%, Very Good=2–5%, Good=5–10%, 10–15%=Average", poor=>15%.

The scores of all indicators for each attribute were summed and divided by the maximum scores to produce a percent score.

$$Score\ Percentage = \frac{Sum\ of\ all\ respondent's\ scores\ for\ each\ indicator}{Max.\ score\ of\ indicator\ *\ No.\ of\ respondents} x\ 100$$

The percent score was used to rank each attribute. The final rank of each attribute was classified as follows: **excellent (attribute score: ≥90%), Very good (attribute score: ≥80% and <90%.), Good (attribute score: ≥60% and <80%), Average (attribute score: ≥50% and <60%) and poor (attribute score: ≤50%).**
The overall attribute score percentage

$$= \frac{Sum\ of\ all\ \ scores\ percentage\ for\ all\ indicator}{No\ of\ Indicators}$$

The overall performance was calculated using below formula

$$overall\ performance = \frac{Sum\ of\ all\ \ Rank\ \ score\ for\ each\ atributes}{No.\ of\ attributes}$$

### Ethical consideration

Permission for this evaluation was obtained from the IEDCR, the EPCBCSP, and the MIS for access to relevant program data. Informed written consent was secured from all interviewees before their participation in the study. To ensure data confidentiality and security, all information was entered into a password-protected Microsoft Excel file to prevent unauthorized access. This evaluation was conducted as part of the academic requirements of the Field Epidemiology Training Program (FETP) in Bangladesh, which is approved by the DGHS under the MOHFW, Bangladesh. Since the evaluation was part of an academic program and did not involve any interventions, Institutional Ethical Review Board (IRB) approval was exempted by the IRB of IEDCR (Memo No. IEDCR/IRB/2024/07).

## 3.0  Result

### Insight from central stakeholders interview

The Project Director of the EPCBCSP and Chief HDU MIS underscores the system's achievements, potential, and areas for improvement. They oversee the surveillance as an initiative aimed at reducing cervical cancer incidence in Bangladesh. The system's core features involve screening of pre-cancerous cervical lesion and CC by VIA and colposcopy, electronic data tracking, real-time monitoring, and data-driven decision-making. Collaboration is facilitated through DHIS2 (MIS), and ministry-level coordination. Notable findings include 18 years of ongoing surveillance, impactful data analysis for policy development, and strategies ensuring data representativeness. The system is expanding coverage, upholding data quality through refresher training, and maintaining privacy measures. Extensive collaboration with healthcare facilities and stakeholders is evident, while advocacy engages healthcare providers and participants. Initiatives like HPV DNA testing and vaccinations stem from findings, with outreach camps capturing data from diverse populations. Challenges encompass limited cancer management facilities and data aggregation concerns. Recommendations involve bolstering data systems, logistics, human resources, government integration, and collaboration with local administrators. The system's strengths encompass electronic data tracking, a population-based approach, and cost-free accessibility. Weaknesses include inadequate staff at surveillance sites and overburdened government staff. Opportunities exist in DHIS2 data accessibility; though specific opportunities aren't outlined.

### Simplicity

The overall score of simplicity was 91.61% (1177/1284), indicating excellent performance. Most indicators of the simplicity attribute received very good to excellent ranks. Notably, 'Case definition is easy to identify a case' and 'Report forms are easy to fill out' received excellent rankings, with scores of 97.5% and 98.5%, respectively. 'Staff received training for cervical cancer surveillance' also had an excellent rank at 95%. 'Follow-up of cases is easy' received a good rank of 63%.

Importantly, there were no unnecessary steps identified in the surveillance system, leading to an excellent rank with a score of 100%. [Table 1]

### Flexibility

The overall score of flexibility was 71.9% (331/460), indicating good performance. Among the flexibility attributes, 'Have there been any changes in the reporting system since the screening center started' and 'Have there been any changes in the information form/registry since the screening center started' received excellent ranks with scores of 95% and 90%, respectively. 'You are adapted to overcome the challenges appeared routinely during system use' had an excellent rank with a score of 99%. 'Have there been any changes in case definition since the screening center started' and 'Does you have access to edit data for wrong update' received good ranks of 60% and 67.5%, respectively. However, 'Does you have access to Change Password of DHIS2' was poorly ranked with a score of 20%. [Table 2]

### Acceptability

The NCCSS demonstrates strong acceptability among its key stakeholders. Participants express a high willingness to engage with the surveillance system, with an excellent rank of 92.5% (185/200) in the statement 'You are willing to participate in this surveillance.' Additionally, the system assures participant privacy and confidentiality effectively, earning an excellent rank of 98.5% (197/200) in the statement 'System assured participant privacy and confidentiality. 'While participants' willingness to participate is considered average, with a rank of 56% (112/200) in the statement 'participants' are willing to participate in this surveillance,' the overall acceptability score remains impressive, achieving a 'Very Good' rating at 85.29% (1054/1234). [Table 3]

### Stability

Within the stability attribute, the NCCSS exhibits notable strengths in some areas, ensuring the reliability and consistency of its operations. Equipment and logistics availability at the screening centers received an excellent ranking of 100%, confirming that these essential resources are consistently provided. However, the system faces some challenges. In the event of outages and power failures, only an average provision of alternate power supply is available, as reflected in a rank of 50%. Furthermore, the absence of dedicated teams for surveillance and poor adaptability to fund variation, with rankings

**Table 1. The simplicity attributes of the NCCSS by score, percent score, and rank in Bangladesh, 2023.**

| Indicators | Score | Score percent | Rank |
|---|---|---|---|
| SOP is easily applicable for operational management of surveillance.[b] | 178 | 89 | Very Good |
| Case definition is easy to identify a case[b] | 195 | 97.5 | Excellent |
| The flow chart/data flow of the system is simple[b] | 174 | 87 | Very Good |
| Report forms are available[a] | 40 | 100 | Excellent |
| Report forms are easy to fill out[b] | 197 | 98.5 | Excellent |
| Data collection is not time consuming[b] | 189 | 94.5 | Excellent |
| Follow-up of cases is easy[b] | 126 | 63 | Good |
| Are there any unnecessary steps in the surveillance system?[a] | 40 | 100 | Excellent |
| Staff received training for Cervical Cancer surveillance[a] | 38 | 95 | Excellent |
| **Overall** | **1177** | **91.61** | **Excellent** |

[a]= Yes, No Responses,

[b]= Likert Scale Responses

**Table 2. The flexibility attributes of the NCCSS by score, percent score, and rank in Bangladesh, 2023.**

| Indicators | Score | Score percent | Rank |
|---|---|---|---|
| Have there been any changes in case definition since the screening center started?[a] | 24 | 60 | Good |
| Have there been any changes in the information form/registry since the screening center started?[a] | 36 | 90 | Excellent |
| Do you have access to edit Data for wrong Update[a] | 27 | 67.5 | Good |
| Do you have access to Change Password of DHIS2[a] | 8 | 20 | Poor |
| Have there been any changes in the reporting system since the screening center started?[a] | 38 | 95 | Excellent |
| You are adapted to overome the challenges appeared routinely during system use.[b] | 198 | 99 | Excellent |
| **Overall** | **331** | **71.9** | **Good** |

[a]= Yes, No Responses,

[b]= Likert Scale Responses

**Table 3. The acceptability attributes of the NCCSS by score, percent score, and rank in Bangladesh, 2023.**

| Indicators | Score | Score percent | Rank |
|---|---|---|---|
| You are willing to participate in this surveillance.[b] | 185 | 92.5 | Excellent |
| You are completely satisfied with the surveillance system.[b] | 162 | 81 | Very Good |
| Participants are willing to participate in this surveillance.[b] | 112 | 56 | Average |
| System assured participant privacy and confidentiality.[b] | 197 | 98.5 | Excellent |
| Are human resources persistently available/posted in this center?[a] | 35 | 87.5 | Very Good |
| Operating experience with existing human resource is Satisfactory[b] | 163 | 81.5 | Very Good |
| A similar system can be consider for future program design[b] | 200 | 100 | Excellent |
| **Overall** | **1054** | **85.29** | **Very Good** |

[a]= Yes, No Responses,

[b]= Likert Scale Responses

of 0% and 76% respectively, indicate areas for potential improvement. Despite these challenges, the overall stability score stands at 53.25%, earning an 'Average' rating **[Table 4]**.

## Usefulness

The NCCSS in Bangladesh is highly effective and valuable. Both of central stakeholder agreed that it is capable of identifying cervical cancer trends in bangladesh. Almost 15 research papers have been published from surveillance activity and seven actions taken by policy makers It consistently generates and disseminates reports, achieving a perfect score of 100%. Moreover, it significantly aids in early prevention and management decisions related to cervical cancer, earning a remarkable score of 99%. Overall, it stands as an indispensable tool, with an impressive 99% score **[Table 5]**.

## Data quality

### Data completeness

The Data Completeness evaluation of the NCCSS in Bangladesh for 2023 revealed varying levels of completeness across different variables. District, NID, Occupation, Patient Visit Date, and Enrollment Date demonstrated "Excellent" completeness, with minimal or no missing data. Conversely, variables such as Education, Family Monthly Income, Husband Living Status, Parity, Union, and Upazila had "Poor" completeness, with substantial missing data. Biopsy, Colposcopic Findings,

**Table 4. The stability attributes of the NCCSS by score, percent score, and rank in Bangladesh, 2023.**

| Indicators | Score | Score percent | Rank |
|---|---|---|---|
| Is equipment and logistics always available for the screening center?[a] | 40 | 100 | Excellent |
| In case of outages and power failures, is alternate power supply available?[a] | 17 | 50 | Average |
| Does surveillance provide computer for data entry?[a] | 0 | 0 | Poor |
| Does program provide internet facility for data entry?[a] | 0 | 0 | Poor |
| Does surveillance provide of referral slip[a] | 40 | 100 | Excellent |
| Availability of core support from surveillance[a] | 40 | 100 | Excellent |
| Availability of dedicated team for surveillance[a] | 0 | 0 | Poor |
| The system could adapt easily to fund variation[a] | 38 | 76 | Good |
| **Overall** | **175** | **53.25** | **Average** |

[a]=Yes, No Responses

**Table 5. The Usefulness of the NCCSS by score, percent score, and rank in Bangladesh, 2023.**

| Indicators | Score | Score percent | Rank |
|---|---|---|---|
| Does surveillance regularly Generate of Report from Surveillance System?[a] | 40 | 100 | Excellent |
| Does surveillance regularly disseminate Report from Surveillance System?[a] | 40 | 100 | Excellent |
| This surveillance is useful for early prevention and management related decision making[a] | 198 | 99 | Excellent |
| How useful is this surveillance system[c] | 198 | 99 | Excellent |
| **Overall** | **436** | **99.33** | **Excellent** |

[a]= Yes, No Responses,

[c]= Usefulness scale (described in methods)

Colposcopic Findings Final, Colposcopy Lesion Size, and Histopathological finding also showed "Poor" completeness. Overall, the system achieved an "Average" (13.11%)completeness score [S1 Table].

## External validity

In order to assess the external validity of this study, a rigorous telephonic communication process was undertaken with a subset of the participant pool. Out of the vast pool of 46383 available phone numbers, a random sample of 100 was meticulously selected. The data obtained from this sample underwent a thorough examination for any potential issues related to validity. It's noteworthy that in this telephonic endeavor, only 15 contacts did not respond to the calls, while 2 individuals declined to answer. The individuals who kindly responded to the calls were queried regarding four pivotal variables: their age, age at marriage, screening results, and union name. The responses received were remarkably accurate, with no discernible validity issues detected during the process.

## Overall performance

The 2023 evaluation of Bangladesh's NCCSS rated the overall performance as "Good" (Rank 3.5). Simplicity and usefulness both received an "Excellent" rating (Rank 5), while acceptability was rated "Very Good" (Rank 4) and flexibility as "Good" (Rank 3). Stability and data quality were both considered "Average" (Rank 2). **[Table 6]**

## Strengths, weaknesses, challenges and recommendations

The system demonstrated several strengths, including early detection of precancerous lesions and cervical cancer, instant report generation, robust support from MIS and program teams, an effective "see and treat" policy, early referrals for treatment, and consistent availability of necessary logistics and equipment. However, the system also faces significant weaknesses, notably the lack of dedicated surveillance staff, limited participant acceptability, and low public awareness of screening importance.

Challenges in the system include motivating participants to undergo screening and managing night shift duties for nurses serving in screening rooms. To overcome these challenges, employing dedicated staff and data entry operators, creating specific nurse positions, providing incentives for staff, enhancing internet connectivity, conducting regular refresher training, ensuring sufficient computers for data entry, motivating participants through physicians, organizing awareness programs and outreach camps, conducting in-center histopathology reports, utilizing media for public awareness, establishing dedicated counseling rooms, and engaging community leaders in awareness efforts. Additionally, there is a recommendation to develop and implement an online self-registration portal that participants can access from home.

## 4.0 Discussion

Periodic evaluations of disease surveillance systems are essential to ensure they meet their objectives and identify areas for improvement. The 2023 evaluation of the NCCSS in Bangladesh presents a nuanced picture of the system's performance. This discussion synthesizes the key findings from the evaluation, reflects on their implications, and provides recommendations for enhancing the system's effectiveness in combating cervical cancer.

The NCCSS has demonstrated several notable strengths. The system excels in simplicity, reflecting its ease of use and the clarity of its case definitions and reporting forms. This high level of simplicity is crucial for effective surveillance, as it ensures that data collection and reporting processes are accessible and manageable for healthcare staff. A study conducted in Congo on influenza surveillance reveals simple surveillance aids to easily capture disease [17]

The usefulness of the NCCSS is noteworthy, achieving a high score of 99%. This indicates that while the system has effectively contributed to early detection and management of cervical cancer, it has room for improvement in assessing specific effectiveness metrics, such as evaluating the impact of screening strategies and identifying high-risk areas. Although the system excels in generating data on cancer trends and influencing policy decisions, further refinement could enhance its role in comprehensive cancer control efforts. A study conducted in Zambia on influenza surveillance reveals similar findings [18].

Flexibility is another critical aspect where the NCCSS performs reasonably well. The system shows adaptability in responding to changes in reporting procedures and integrating new data sources. However, issues such as variability in funding and limited access to essential features like password changes may hinder overall flexibility. Enhancing these

**Table 6. The overall performance of the NCCSS in Bangladesh, 2023.**

| Indicators | Score | Percent score | Rank score | Rank* |
|---|---|---|---|---|
| Simplicity | 1177 | 91.61 | 5 | Excellent |
| Flexibility | 331 | 71.9 | 3 | Good |
| Acceptability | 1054 | 85.29 | 4 | Very Good |
| Stability | 175 | 53.25 | 2 | Average |
| Usefulness | 436 | 99.33 | 5 | Excellent |
| Data quality | NA | 13.11 | 2 | Average |
| Overall performance | | | 3.5 | **Good** |

*Excellent =5, Very Good=4. Good=3, Average=2, poor=1

aspects could improve the system's ability to adapt to evolving needs and challenges. Similarly studies conducted on Measles, acute flaccid paralysis and National Tetanus Surveillance system (NTSS) found flexible to incorporating new elements in the surveillance [19,20].

Despite its strengths, the NCCSS faces several challenges. The evaluation revealed stability concerns, with an average score. While the system effectively manages equipment and logistics, challenges such as inadequate backup power and the absence of dedicated surveillance teams reflect its instability. These issues undermine the system's reliability and sustainability, suggesting a need for improvements in resource allocation and infrastructure support. Similar findings werefound in NTSS, while Severe Acute Respiratory Illness Sentinel Surveillance System in Yemen shows good stability contrast our finding [20,21].

Data quality emerged as a significant area for improvement, with an overall completeness score reflecting average performance. Variables such as education and biopsy results showed poor completeness, which could affect the accuracy and reliability of data analysis. Addressing these gaps through enhanced data collection protocols and staff training is crucial for improving the system's overall effectiveness. Low-quality data can greatly hinder its usefulness for planning and making informed decisions. This problem may stem from differences in how health workers report data and their varying levels of commitment to surveillance activities. When data is not consistently accurate, it can undermine public health efforts and decision-making processes [22]. A study conducted in Brazil emphasized health managers should prioritize enhancing data quality and establishing continuous evaluation routines to improve screening effectiveness [23].

Acceptability of the NCCSS is strong, with an impressive score of 92.5%. The high willingness of participants to engage with the system is a positive indicator of its perceived value. Nevertheless, the average satisfaction level among some stakeholder's points to potential communication gaps and areas for improved interaction between central and local levels.

The system effectively detects precancerous lesions and cervical cancer, with strong support and logistics. However, it lacks dedicated surveillance staff, faces participant acceptability issues, and suffers from low public awareness. Challenges include participant motivation and night shift management. Solutions include hiring dedicated staff, improving connectivity, offering incentives, enhancing training, increasing public awareness, and creating an online self-registration portal. Similarly, the VIA-based screening program in Morocco showed improvements in provider training, health information systems, community awareness, and a single-visit approach was required for managing lesions [24].

## Limitation

This evaluation has some limitations. Sensitivity and positive predictive value (PPV) could not be calculated because only VIA-positive cases were referred for colposcopy, leaving data on false negatives and false positives incomplete. Additionally, representativeness was not assessed due to the lack of comprehensive data from all geographic areas and healthcare settings. Furthermore, only a few sites were included in the study, which may limit the generalizability of the findings.

## Conclusion

The 2023 evaluation of Bangladesh's National Cervical Cancer Surveillance System (NCCSS) underscores its strengths in early detection and its simplicity and flexibility in operation. However, the evaluation also reveals significant limitations, including challenges with system stability, data quality, and public awareness. To enhance the effectiveness of the NCCSS, it is recommended to improve infrastructure and data management protocols, as well as implement targeted strategies to increase public engagement and awareness. Additionally, the introduction of an online self-registration portal could streamline participant interaction and improve system efficiency. While the NCCSS demonstrates potential in cervical cancer control, addressing these limitations is crucial for strengthening its surveillance capacity and ensuring its long-term sustainability and impact.

## Supporting information

**S1 Table.  The data completeness of the NCCSS in Bangladesh, 2023.**
(DOCX)

**S1 Text.  Questionnaire used for interview.**
(PDF)

**S1 Data.  Raw anonymized dataset used in the analysis.**
(XLSX)

## Acknowledgments

The author wishes to acknowledge the Institute of Epidemiology, Disease Control and Research (IEDCR) and the South Asia Field Epidemiology and Technology Network (SAFETYNET) for their support in this evaluation. We also grateful to Gretchen Cowman (Epidemiologist) from Center for disease control United states for her tremendous support throughout the evaluation.

## Author contributions

**Conceptualization:** Md Foyjul Islam, Quazi Ahmed Zaki, Tahmina Shirin.

**Data curation:** Md Foyjul Islam.

**Formal analysis:** Md Foyjul Islam.

**Funding acquisition:** Md Foyjul Islam, Md. Omar Qayum, Mohammad Rashedul Hassan, Tahmina Shirin.

**Investigation:** Md Foyjul Islam.

**Methodology:** Md Foyjul Islam, Quazi Ahmed Zaki, Shah Ali Akbar Ashrafi.

**Project administration:** Md Foyjul Islam.

**Resources:** Md Foyjul Islam, Shah Ali Akbar Ashrafi.

**Software:** Md Foyjul Islam.

**Supervision:** Md Foyjul Islam, Ashrafun Nessa, Quazi Ahmed Zaki, Md. Omar Qayum, Mohammad Rashedul Hassan, Tahmina Shirin.

**Validation:** Md Foyjul Islam, Ashrafun Nessa, Quazi Ahmed Zaki, Tahmina Shirin.

**Visualization:** Md Foyjul Islam, Quazi Ahmed Zaki.

**Writing – original draft:** Md Foyjul Islam, Ashrafun Nessa, Quazi Ahmed Zaki, Shah Ali Akbar Ashrafi, Md. Omar Qayum, Mohammad Rashedul Hassan, Tahmina Shirin.

**Writing – review & editing:** Md Foyjul Islam, Ashrafun Nessa, Quazi Ahmed Zaki, Shah Ali Akbar Ashrafi, Md. Omar Qayum, Mohammad Rashedul Hassan, Tahmina Shirin.

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
