## [Decision Letter · Decision Letter 0]

5 Feb 2025

PGPH-D-24-02609

Evaluation of the National Cervical Cancer Surveillance in Bangladesh: Performance, Strengths, and Opportunities for Improvement

Dear Dr. Islam,

Thank you for submitting your manuscript to PLOS Global Public Health. After careful consideration, we feel that it has merit but does not fully meet PLOS Global Public Health’s publication criteria as it currently stands. Therefore, we invite you to submit a revised version of the manuscript that addresses the points raised during the review process.

We look forward to receiving your revised manuscript.

Kind regards,

Vanessa Carels

Staff Editor

Journal Requirements:

**Please only choose the relevant sentences from below**

i. Please clarify all sources of funding (financial or material support) for your study. List the grants (with grant number) or organizations (with url) that supported your study, including funding received from your institution. 

ii. State the initials, alongside each funding source, of each author to receive each grant.

iii. State what role the funders took in the study. If the funders had no role in your study, please state: “The funders had no role in study design, data collection and analysis, decision to publish, or preparation of the manuscript.”

iv. If any authors received a salary from any of your funders, please state which authors and which funders.

2. Please make sure the funding information on the submission form matches your financial disclosure statement. Please indicate by return the full and correct funding information for your study and confirm the order in which funding contributions should appear. Please be sure to indicate whether the funders played any role in the study design, data collection and analysis, decision to publish, or preparation of the manuscript.

3. We have amended your Competing Interest statement to comply with journal style. We kindly ask that you double check the statement and let us know if anything is incorrect. 

4. In the online submission form, you indicated that "The data collected for this study has been analyzed and is included

within this published article. The data analyzed in this study is available on justifiable request.". 

3. Uploaded as supplementary information.

5. We have noticed that you have uploaded Supporting Information files, but you have not included a list of legends. Please add a full list of legends for your Supporting Information files after the references list. 

6. We notice that your supplementary table 1 has a manuscript file. Please remove them and upload them with the file type 'Supporting Information'. Please ensure that each Supporting Information file has a legend listed in the manuscript after the references list.

7. Figure 2: please (a) provide a direct link to the base layer of the map (i.e., the country or region border shape) and ensure this is also included in the figure legend; and (b) provide a link to the terms of use / license information for the base layer image or shapefile. We cannot publish proprietary or copyrighted maps (e.g. Google Maps, Mapquest) and the terms of use for your map base layer must be compatible with our CC-BY 4.0 license. 

Additional Editor Comments (if provided):

Reviewers' comments:

Reviewer's Responses to Questions

**Comments to the Author**

1. Does this manuscript meet PLOS Global Public Health’s publication criteria ? Is the manuscript technically sound, and do the data support the conclusions? The manuscript must describe methodologically and ethically rigorous research with conclusions that are appropriately drawn based on the data presented.

Reviewer #1: Yes

Reviewer #2: Yes

Reviewer #3: Yes

2. Has the statistical analysis been performed appropriately and rigorously?

Reviewer #1: N/A

Reviewer #2: Yes

Reviewer #3: I don't know

3. Have the authors made all data underlying the findings in their manuscript fully available (please refer to the Data Availability Statement at the start of the manuscript PDF file)?

Reviewer #1: Yes

Reviewer #2: Yes

Reviewer #3: No

4. Is the manuscript presented in an intelligible fashion and written in standard English?

Reviewer #1: No

Reviewer #2: Yes

Reviewer #3: No

5. Review Comments to the Author

Reviewer #1: Presented study evaluates the performance of the Electronic Data Tracking with Population-Based Cervical and Breast Cancer Screening Program (EPCBCSP) in Bangladesh, aimed at addressing cervical cancer (CC), which is highly prevalent in the country. Data from 42 stakeholders across four divisions were analyzed using CDC guidelines, focusing on system attributes such as simplicity, flexibility, acceptability, stability, and usefulness. The overall performance score was 77.2%, with simplicity and usefulness rated as excellent, and acceptability as very good. Challenges included staff shortages and participant motivation. The study highlights the need to improve system stability and data quality to strengthen the program's effectiveness in reducing CC burden. The following are some major concerns with the manuscript:

• Abstract background does not identify the research gap in the field, rather elaborates how the research study was conducted. It is important to mention why the research was conducted and how it bridges the gap of exciting knowledge in the field.

• It is unusual to have subheading for different part of the Introduction section. It should have a flow of background knowledge, present condition and treatments of cervical cancer in Bangladesh, current research gap and finally what the current study aims, objectives and brief results summary without any subheading.

• Methods section should detail as much as possible. Since the data were collected via in depth interview using a semi-structured questionnaire adapted from updated CDC guideline, a copy of such questioner should be included in the supplementary document for the evaluation.

• Although, authors advised no ethical approval needed, it is highly recommended to cite the ethical approval reference of original patients’ data collection.

• No statistical analysis of the data was done. Please select an appropriate statistical analysis such as confidence interval for the data.

• Results section must highlight result-oriented subheadings and section rather than adjective titles.

• Similar to Introduction section, the Discussion section should not have any subheadings or highlights.

• Conclusion section should concisely summarize the Bangladesh's NCCSS and it’s limitations.

• English throughout the manuscript is erroneous and thus difficult to understand.

Reviewer #2: Minor revisions

Introduction

CC is not an acceptable synonym for cervical cancer.

Authors begin the sentence (Page 3, Line No: 82) as CC, which is never acceptable.

Authors have to get Ethical clearance from Institutional Research Board.

Ethical clearance Number should be mentioned.

Authors should add limitations of VIA

They should discuss the latest recommended method for cervical cancer screening (HPV DNA testing)

Reviewer #3: Thank you for inviting me to review this important and relevant manuscript and analysis evaluating the Electronic Data Tracking with Population Based Cervical and Breast

Cancer Screening Program (EPCBCSP) in Bangladesh

In general, it needs to be clear from title that this is qualitative evaluation of the surveillance systems through interviews with key stakeholders involved in the surveillance programme, and not an evaluation of the cervical cancer screening and treatment programme itself. The current title is a bit misleading.

In general, the presentation of results needs some simplification. Currently, authors are assuming that the reader has a detailed understanding of the CDC methodology for evaluating surveillance . More work is needed to explain in methods, the individual outcomes evaluated, and the results section need to detail what the numerators and denominators refer to.

Specific comments:

In methods, authors refer to “CDC guideline for evaluating public health surveillance system” Authors should provide definitions for each of the outcomes reported : simplicity, flexibility, acceptability, and stability, and usefulness. How were each of these outcomes defined and measured? For example: the outcome of acceptability - is this referring to acceptability of screening to the woman undergoing screening or the healthcare provider providing the service? Or is it acceptability of the surveillance system at healthcare provider level?

Further information is also needed on how the attribute is qualified, i.e. what is the gold standard for that particular attribute (how is the maximum score assigned? what does maximum score look like? )

As it was difficult to understand what attributes of the cervical cancer screening programme are being evaluated, it was difficult to interpret the findings reported.

Line 226 – The numerator and denominator ((1177/1284) need to be explained. What do these numbers refer to ? Individuals interviewed, clinics?

Lin 228: “Case definition is easy to identify a case' . Does a case refer to cervical cancer ,or cervical precancerous lesions?

Line 230: 'Follow-up of cases is easy – again is this follow-up of women with cervical precancer or cervical cancer. How a case is defined is important because the ease of follow-up will be very different depending on whether it is precancer or cancer

Table 1 – it is not clear how the column “score” is quantified. How Is “score percent” calculated?

How does it relate to the earlier figures in line 226 (1177/1284) ?

Table 1- footnote b refers to Likert scale responses, but this is not explained in the methods

A more detailed description of the stakeholders interviewed would be helpful – does it include individuals actively involved in the surveillance of screening and treatment? In what capacity? (number/% health care provider , data cleaning/analysis, interpretation, high level stakeholders etc.)

Minor comments

Can authors confirm the sentence “ All ever-married women over the age of 30 made

up the target group for CC screening,. What of unmarried women over the age of 30 years? I refer to the National Strategy for Cervical Cancer Prevention (2017-2022) which states that “The target population for screening will be women between 30-60 years of age; and the screening interval will be 5 years. “

page 4, line 112: there are too many abbreviations, suggest simplifying for ease of reading “specialist group represented by DGHS, 112 DGFP, NICRH, BSMMU, OGSB, WHO, UNFPA, UNICEF and approved by MOHFW [12]”

line 119: could authors clarify the level of health facility referred to “across 601 health facilities covering all subdistricts in Bangladesh”, i..e what population do these 601 health facilities represent – urban, semi-urban, rural?

Lines 206-224: It’s not clear if this paragraph details a finding from the interviews or background information on the screening and surveillance programme. If the latter, it should be given in the introduction

6. PLOS authors have the option to publish the peer review history of their article (what does this mean? ). If published, this will include your full peer review and any attached files.

**Do you want your identity to be public for this peer review?** For information about this choice, including consent withdrawal, please see our Privacy Policy .

Reviewer #1: No

Reviewer #2: **Yes: ** Dr Sabeena Sasidharan Pillai

Reviewer #3: No

---

## [Decision Letter · Decision Letter 1]

15 Apr 2025

Evaluation of the National Cervical Cancer Surveillance Program in Bangladesh: Performance, Strengths, and Opportunities for Improvement

PGPH-D-24-02609R1

Dear Dr Islam,

We are pleased to inform you that your manuscript 'Evaluation of the National Cervical Cancer Surveillance Program in Bangladesh: Performance, Strengths, and Opportunities for Improvement' has been provisionally accepted for publication in PLOS Global Public Health.

Best regards,

Julia Robinson

Executive Editor

Reviewer Comments (if any, and for reference):

Reviewer's Responses to Questions

**Comments to the Author**

1. If the authors have adequately addressed your comments raised in a previous round of review and you feel that this manuscript is now acceptable for publication, you may indicate that here to bypass the “Comments to the Author” section, enter your conflict of interest statement in the “Confidential to Editor” section, and submit your "Accept" recommendation.

Reviewer #2: All comments have been addressed

2. Does this manuscript meet PLOS Global Public Health’s publication criteria ? Is the manuscript technically sound, and do the data support the conclusions? The manuscript must describe methodologically and ethically rigorous research with conclusions that are appropriately drawn based on the data presented.

Reviewer #2: Yes

3. Has the statistical analysis been performed appropriately and rigorously?

Reviewer #2: Yes

4. Have the authors made all data underlying the findings in their manuscript fully available (please refer to the Data Availability Statement at the start of the manuscript PDF file)?

Reviewer #2: Yes

5. Is the manuscript presented in an intelligible fashion and written in standard English?

Reviewer #2: Yes

6. Review Comments to the Author

Reviewer #2: article can be accepted.

7. PLOS authors have the option to publish the peer review history of their article (what does this mean? ). If published, this will include your full peer review and any attached files.

**Do you want your identity to be public for this peer review?** For information about this choice, including consent withdrawal, please see our Privacy Policy .

Reviewer #2: **Yes: ** Sabeena Sasidharan Pillai
